# Late Cenozoic unification of East and West Antarctica

Roi Granot[1] & Jérôme Dyment [2]

The kinematic evolution of the West Antarctic rift system has important consequences for regional and global geodynamics. However, due to the lack of Neogene seafloor spreading at the plate boundary and despite being poorly resolved, East-West Antarctic motion was assumed to have ended abruptly at 26 million years ago. Here we present marine magnetic data collected near the northern edge of the rift system showing that motion between East and West Antarctica lasted until the middle Neogene (~11 million years ago), long after the cessation of the known mid-Cenozoic pulse of motion. We calculate new rotation parameters for the early Neogene that provide the kinematic framework to understand the varied lithospheric settings of the Transantarctic Mountains and the tectono–volcanic activity within the rift. Incorporation of the Antarctic plate motion into the global plate circuit has major implications for the predicted Neogene motion of the Pacific Plate relative to the rest of the plates.

[1] Department of Geological and Environmental Sciences, Ben-Gurion University of the Negev, Beer-Sheva 84105, Israel. [2] Institut de Physique du Globe de Paris, CNRS UMR 7154, Sorbonne Paris Cité, Université Paris Diderot, Paris 75005, France. Correspondence and requests for materials should be addressed to R.G. (email: rgranot@bgu.ac.il)

The West Antarctic rift system (WARS), dividing the East and West Antarctic Plates (Fig. 1) has undergone repeated rifting episodes since its creation during the breakup of Gondwana in the Mesozoic[1,2]. The kinematic history of the rift has played critical roles in shaping the structure of the lithosphere and the thermal state of Antarctica[3–5], in the formation of sub-ice topography[6] and in the uplift of the Transantarctic Mountains (TAM)[7]. Moreover, Antarctica forms an important link between the Pacific and Indo-Atlantic realms in the global plate tectonic circuits. Understanding past plate motions between East and West Antarctica, therefore, dictates our ability to accurately predict the kinematic evolutions of other plate boundaries. Yet our knowledge of the relative plate motions of East and West Antarctica is only well constrained for the Eocene and Oligocene periods, between 43 and 26 million years (Myr) ago, when a pulse of rifting led to ultraslow seafloor spreading at the northwestern end of the rift system[8]. Although high heat flow[4,5] and volcanic activity[9] still persist throughout large parts of the rift system, minimal tectonic-related seismic activity[10] coupled with negligible geodetically-constrained relative plate motion between East and West Antarctica[11] indicate that the rift is no longer kinematically active. The nature of the transition from the Eocene-Oligocene rifting phase to the unification of East and West Antarctica into a coherent plate, however, is unclear and was assumed to have happened abruptly at 26 Myr ago, leading to inconsistencies in previous regional and global plate reconstructions[12,13].

Since the creation of the WARS, motions between the two plates have shifted and concentrated along the western Ross Sea embayment and at the front of the TAM[8,14–16]. During the Eocene-Oligocene period, when continental rifting evolved into ultraslow seafloor spreading in the Adare Basin[17–19], the WARS was one arm of a northward-drifting ridge-ridge-ridge triple junction. That meeting of the three plate boundaries (Australia-East Antarctica and Australia-West Antarctica along the Southeast Indian Ridge (SEIR), and East-West Antarctica along the WARS, Fig. 1) allowed their magnetic anomalies, fracture zones and other independent kinematic constraints to simultaneously constrain the rotation parameters for East-West Antarctica motion[8,18,20]. Geological and geophysical observations suggest that prominent Neogene deformation that post-dates the Adare seafloor spreading has been accommodated along large parts of the rift system. Pronounced lower and middle Miocene normal faulting created a series of tilted blocks within the sedimentary sequence deposited on top of the Oligocene oceanic crust of the western Adare Basin[21] (Fig. 1).

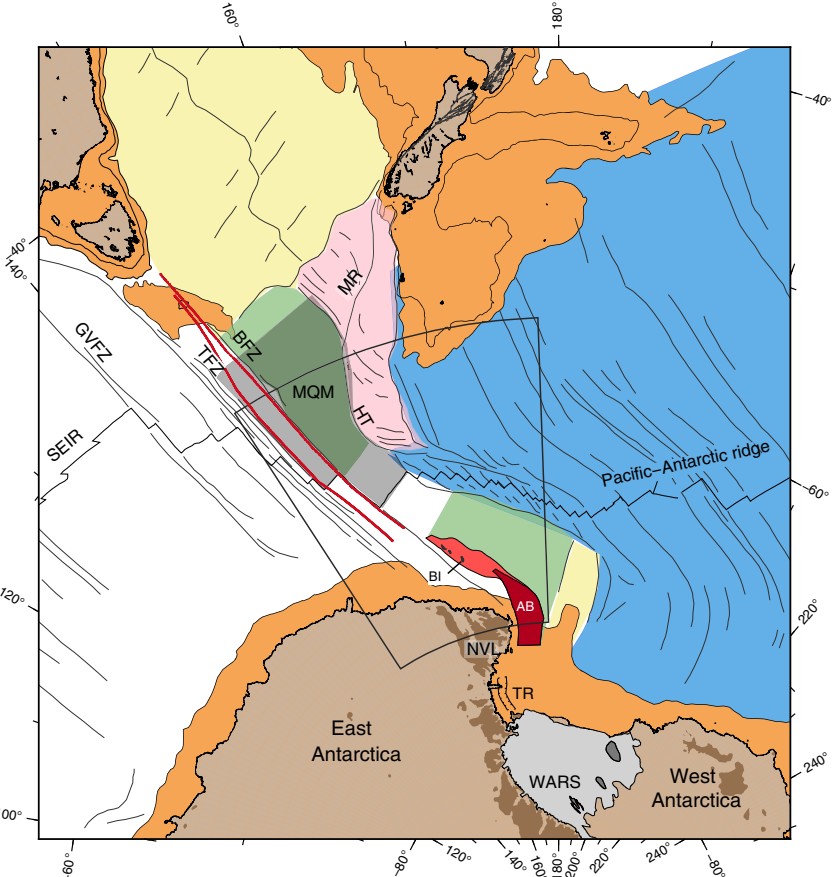

**Fig. 1** Tectonic map of the southwest Pacific and southeast Indian Oceans. The white region was generated by Australia-East Antarctica (and, after 11 Myr, Australia-Antarctica) seafloor spreading, and the light green area was generated by Australia-West Antarctica spreading. The grey-shaded area north of the southeast Indian Ridge (SEIR) marks the location of the Macquarie microplate (MQM). The dark red area was generated by East-West Antarctic spreading (between anomalies 20o and 8o) and the light red region marks the area affected by East-West Antarctica motion between anomalies 8o and 5o. Red lines delineate the track of the IB L'Astrolabe TACT cruises. Light blue, yellow and pink areas were generated by Pacific-West Antarctica (and, after 11 Myr, Pacific-Antarctica), Australia-Lord Howe Rise and Australia-Pacific spreading, respectively. WARS, West Antarctic rift system; BI, Balleny Islands; AB, Adare Basin; TR, Terror Rift; GVFZ, TFZ and BFZ are the George V, Tasman and Balleny fracture zones, respectively; NVL, Northern Victoria Land; TAM, Transantarctic Mountains; HT, Hjort Trench; MR, Macquarie ridge. The grey annular outline shows the location of Fig. 2

Concurrently, transtensional faulting and volcanic activity occurred onshore in Northern Victoria Land[22,23]. Further south, the offshore Terror Rift developed and accommodated 10 to 15 km of east-west opening[24] with an overall northwest-southeast sense of motion[25]. Findings of thin sedimentary infill[26] and low effective elastic thickness[27] in the basins and troughs beneath the West Antarctic Ice Sheet suggest that the crust there also underwent deformation during the Neogene[28]. If the rift indeed acted as a plate boundary during the Neogene, evidence of its motion should be documented in the oceanic crust younger than 26 Myr that formed along the two other arms of the triple junction (i.e., Australia-East Antarctica and Australia-West Antarctica spreading systems, Fig. 1).

We acquired marine magnetic data along the two edges of the Tasman spreading corridor located between the Tasman and Balleny fracture zones (Fig. 1) during two cruises of IB L'Astrolabe. Together with archival well-navigated data acquired from the area located east of the Balleny fracture zone and west of the Hjort Trench (Balleny spreading corridor), these data enable us to investigate the kinematics of seafloor spreading east and west of the WARS and to calculate the rotation parameters for Neogene motion between East and West Antarctica.

## Results and Discussion

**East-West Antarctic plate motion**. If there was an active plate boundary in Antarctica during the Neogene, then indications of East-West Antarctic motion should be apparent when rotating the Australian anomalies that are younger than anomaly 8o (26.0 Myr, anomaly ages are from Ogg[29]) back to East Antarctica by using the rotation parameters calculated based on data from west of the George V fracture zone (Fig. 1). Matching between the Antarctic 8o and younger isochrons in both the Tasman and Balleny corridors would indicate that the WARS was no longer active during the Neogene. However, formation of the Macquarie microplate some 6 Myr ago[30,31] north of the SEIR and east of the Tasman fracture zone[32] (Fig. 1) complicates this task. The recent clockwise rotation of the microplate affected the locations of the Australian magnetic anomalies and had to be corrected for before these

isochrons could be rotated to Antarctica. Therefore, first we recalculated the Macquarie-Australia rotation parameters for anomaly 3Ay (6 Myr) using the archival and new magnetic constraints (see Methods, Supplementary Figs. 1–3 and Table 1 for details). During reconstruction of the isochrons (3Ay, 5o, 6y and 8o) found north of the ridge axis, first to Australia and then to Antarctica by using the Macquarie-Australia and Australia-East Antarctica rotations, respectively (Fig. 2), we observed an excellent fit for both anomalies 3Ay and 5o. Most importantly, this fit is observed on the crust that confines the northern edge of the WARS, on both the Tasman and Balleny corridors. This observation indicates that it is unlikely that significant relative plate motion took place between East and West Antarctica after anomaly 5o (11 Myr).

Unlike the young isochrons, the rotated anomalies 6y and 8o (18.7 and 26 Myr, respectively) from the Australian flank exhibit good matches with their conjugates on the Antarctic flank in the Tasman corridor but mismatches with those in the Balleny corridor (Fig. 2). These anomalies misfit the positions of their Antarctic counterparts by about 45 km (anomaly 6y) and 60 km (anomaly 8o). Interestingly, regardless of where the Macquarie-Australia Euler pole is positioned, it could not bring the Macquarie isochrons found west and east of the Balleny fracture zone into simultaneous fit with their conjugate Antarctic isochrons. The observed increasing misfit found along the Balleny corridor for anomalies 5o, 6y and 8o could be the result of either complex distributed internal deformation within the Balleny corridor of the Macquarie microplate or, alternatively, the existence of a third-plate boundary located south of the SEIR axis and between the Tasman and Balleny corridor anomalies. Taken together, the fracture zones and isochron orientations and the seismicity pattern suggest that since its creation, the Macquarie microplate extended from the Tasman fracture zone on the west to the Hjort Trench on the east[30–32] (Fig. 1). The absence of pronounced internal deformation in a seismic reflection profile acquired across the microplate (Supplementary Fig. 4) and the lack of any abnormal gravity feature north of the ridge axis (Fig. 2) suggest that the Macquarie microplate could not have accommodated sufficient internal deformation to explain

### Table 1 Finite rotations and their covariance matrices

| Mag. Ano. | Age (Myr) | Lat. (°N) | Long. (°E) | Angle (°) | $\hat{K}$ | a | b | c | d | e | f | g | Points | Segs. |
|---|---|---|---|---|---|---|---|---|---|---|---|---|---|---|
| | | | | Macquarie-Antarctica | | | | | | | | | | |
| 3Ay | 6.0 | −43.18 | −169.14 | 7.55 | 2.28 | 2.28 | −1.19 | 4.14 | 0.63 | −2.15 | 7.57 | 5 | 32 | 4 |
| | | | | Macquarie-Australia | | | | | | | | | | |
| 3Ay | 6.0 | −54.57 | 150.63 | 5.28 | 2.15 | 2.31 | −1.23 | 4.17 | 0.69 | −2.20 | 7.66 | 5 | N/A | N/A |
| | | | | Australia-East Antarctica | | | | | | | | | | |
| 8o | 26.0 | −13.79 | −146.43 | 15.92 | 0.95 | 1.68 | −2.05 | 1.05 | 3.57 | −2.72 | 5.04 | 7 | 137 | 18 |
| | | | | Australia-West Antarctica | | | | | | | | | | |
| 8o | 26.0 | −11.88 | −145.66 | 15.98 | 0.95 | 2.59 | −0.66 | 5.53 | 0.20 | −1.40 | 11.90 | 5 | 137 | 18 |
| | | | | East Antarctica-West Antarctica | | | | | | | | | | |
| 8o | 26.0 | 67.05 | −109.47 | 0.57 | 0.95 | 1.14 | 0.34 | 3.89 | 0.13 | 1.10 | 13.37 | 5 | 137 | 18 |

Rotations for the plate pairs are given as motion of the first plate relative to the second. The covariance matrix is given by the formula $\frac{1}{k} * \begin{pmatrix} a & b & c \\ b & d & e \\ c & e & f \end{pmatrix} \times 10^{-g}$ radians$^2$. Points (Segs.) are the total number of magnetic anomaly and fracture zone points (segments) used in the calculation

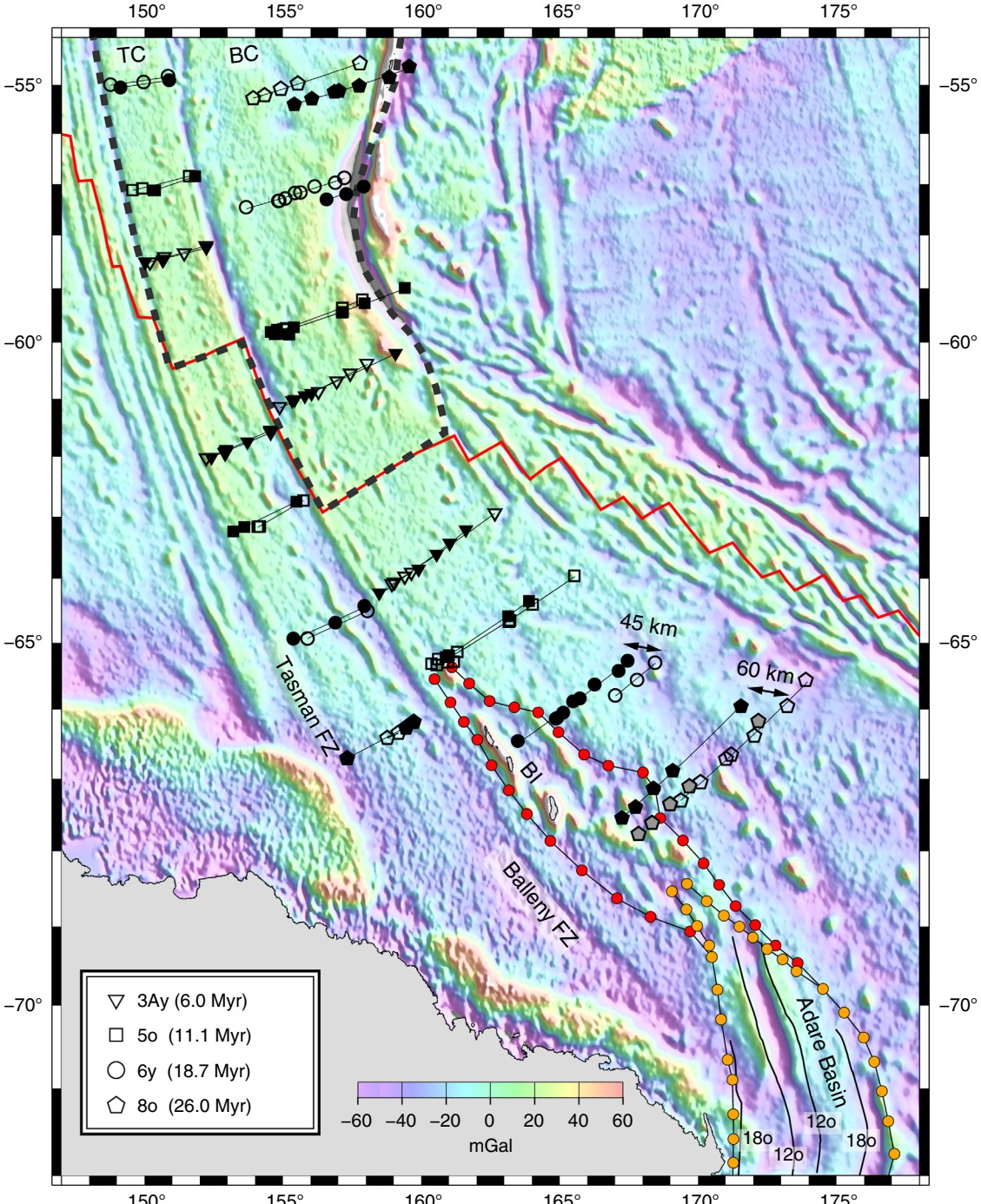

**Fig. 2** Magnetic anomaly locations in the Balleny and Tasman corridors. Background is the satellite-derived free-air gravity field[53]. The positions of the anomalies (open symbols) are compared to those from the conjugate plate (filled symbols), rotated using the Australia-East Antarctica rotations[30]. Anomalies from the Macquarie microplate, delineated by a dashed line, are corrected for Macquarie-Australia motion (Table 1). Rotated anomalies 6y and 8o from the Balleny corridor misfit the positions of their Antarctic counterparts by 45 and 60 km, respectively. Grey-shaded anomaly picks (8o locations) are rotated using the Macquarie-Australia-West Antarctica rotation (Table 1). Dotted red lines delineate the region of crust affected by Neogene East-West Antarctic motion. Dotted orange lines confine the Adare Basin with its internal anomalies 12o and 18o. BI, Balleny Islands; TC, Tasman corridor; BC, Balleny corridor

the mismatch observed for anomalies 6y and 8o. Therefore, the increase in the level of mismatch with age found in the older anomalies east of the Balleny fracture zone suggests that the Tasman and Balleny corridors record two different kinematic histories and were therefore divided by a third-plate boundary.

A strand of linear and elongated gravity features (highlighted by red circles in Fig. 2), possibly reflecting a series of linear faults and volcanic edifices[33], extends along the western side of the Balleny corridor, from the northern edge of the Adare Basin to the Antarctic anomaly 5o. This 500-km-long feature is located between the two sets of isochrons that are part of the Australia-

East Antarctica and Australia-West Antarctica spreading systems. It is not matched by a similar feature on the conjugate crust north of the SEIR axis, suggesting that this area was deformed after it formed between 26 and 11 Myr ago, and therefore, it likely preserves the northernmost end of the WARS. This feature includes the Balleny Islands, part of the still active Balleny volcanic province, the formation of which was previously ascribed to a mantle hotspot plume[34]. Its geochemical signature, similar to that of other volcanic provinces of the WARS[33], and the lack of seamount trail suggest that the origin of the Balleny volcanic province might have been related to the extensional motion accommodated along the WARS. The apparent lack of deformation in the southern Balleny fracture zone, found immediately west of this area (Fig. 2), indicates that little, if any, East-West Antarctic motion has been accommodated by the dormant Balleny fracture zone.

The cumulating evidence for early to middle Miocene displacements along the rift system, from the northern Balleny Islands toward the western Adare Basin, Northern Victoria Land and Terror Rift, suggests that a common plate boundary was active in the western Ross Sea area during the early Neogene. Using the plate circuit linking East Antarctica to Australia to West Antarctica, the rotation parameters that depict the East-West Antarctica relative plate motions between anomalies 8o and 5o can be calculated. The magnetic anomalies and fracture zones of the SEIR found west of the George V fracture zone constrain the Australia-East Antarctica motions, those east of the Balleny fracture zone constrain the Australia-West Antarctica motions, and, assuming that it reflects the total motion younger than 26 Myr accommodated at that latitude, the extension at the Terror Rift constrains the East-West Antarctic motions (Fig. 3). We simultaneously computed the rotation parameters and their uncertainties[35,36] for the three-plate boundaries at the time of anomaly 8o. We assigned 12.5 km of opening to the Terror Rift to reflect the seismically-measured total extension of 10 to 15 km there[24]. The data used and the calculated kinematic solution are shown in Fig. 4. The best-fit rotation parameters for the three-plate boundaries are given in Table 1. Rotating the Australian anomaly 8o picks from the Balleny corridor by the Macquarie-Australia-West Antarctica rotations eliminates the misfit of the 8o magnetic picks (Fig. 2). We note that varying the opening values used for the Terror Rift between 5 and 20 km does not significantly change the rotation parameters for East-West Antarctica. Larger opening values predict an unrealistic extension in the Adare Basin and Balleny Islands sector. Finally, our currently insufficient knowledge about how extension has been accommodated by the Terror Rift over time precludes division of the East-West Antarctic motion of 26 to 11 Myr ago into several stages.

The calculated Euler pole describing the East-West Antarctic motion since the time of anomaly 8o and until the cessation of rifting 11 Myr ago has a rather large 95% confidence region due to the relatively small spatial extent of the Terror Rift, which constrains the Antarctic plate boundary motion (Fig. 4), regardless of the quality of the constraining data. Nevertheless, the new pole provides important insights on the development of the rift system. Although the opening direction of the Terror Rift has not been imposed on the kinematic solution, only its east-west displacement, the solution predicts a general trend of NW-SE sense of motion there (Fig. 5). This predicted direction agrees, albeit with large uncertainty, with the opening directions inferred directly from investigation of calcite twinning in the ANDRILL AND-1B core[25]. According to the new rotation parameters, the predicted extensional motion that has taken place in the Adare Basin during the Neogene was $12 \pm 6$ km ($1\sigma$), which stands in general agreement with the observed ~7 km of extension

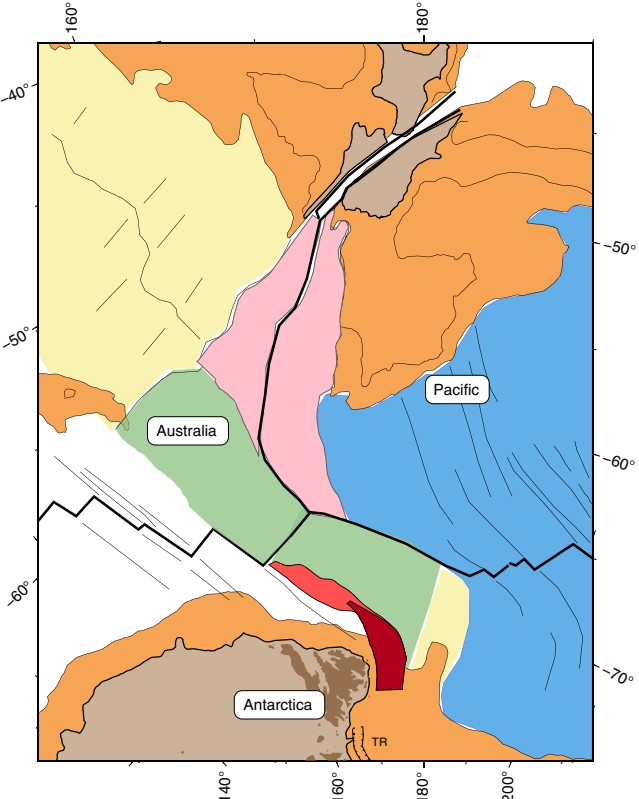

**Fig. 3** Plate reconstruction for the time of anomaly 5o, shortly after the final cessation of East-West Antarctic plate motion. Antarctica is fixed. The more recent displacement caused by the formation of the Macquarie microplate is restored using the Macquarie-Australia rotation parameters (Table 1). Thick black lines delineate active plate boundaries. Colour shadings follow those in Fig. 1. TR, Terror Rift

calculated by the summation of offsets across the seismically imaged faults[21].

**Implications for the West Antarctic rift system.** The late Cenozoic displacement field within the WARS (Fig. 5) has varied from extensional-dominated motion in the Ross Sea sector to oblique convergence motion beneath the West Antarctic Ice Sheet. The transition between these two contrasting regimes of motion is predicted to have happened under the Ross Ice Shelf, where negligible ($0 \pm 12$ km) displacement orthogonal to the front of the Transantarctic Mountains was accommodated during the Neogene. The contrasting kinematic behaviour predicted for the two edges of the rift system may have affected the mechanism by which the different parts of the TAM have been uplifted. Seismic imaging studies have found contrasting crustal and mantle architectures along the mountain chain. The narrow northern TAM (Fig. 5) was suggested to have formed by flexural bending of a thick lithosphere[37], supported by a thermally buoyant upper mantle associated with extension within the rift[38]. In the central TAM, on the other hand, no thermal anomaly is observed[39] and only flexural bending is invoked as the mechanism for mountain formation. In contrast, a recent seismic tomography study[40] showed that the wide southern TAM is undergoing lithospheric foundering, whereby the lower lithosphere is sinking into the mantle. We suggest that the contrasting senses of motion between 43 and 11 Myr might have influenced the mechanisms by which the different parts of the TAM have been uplifted, resulting in the

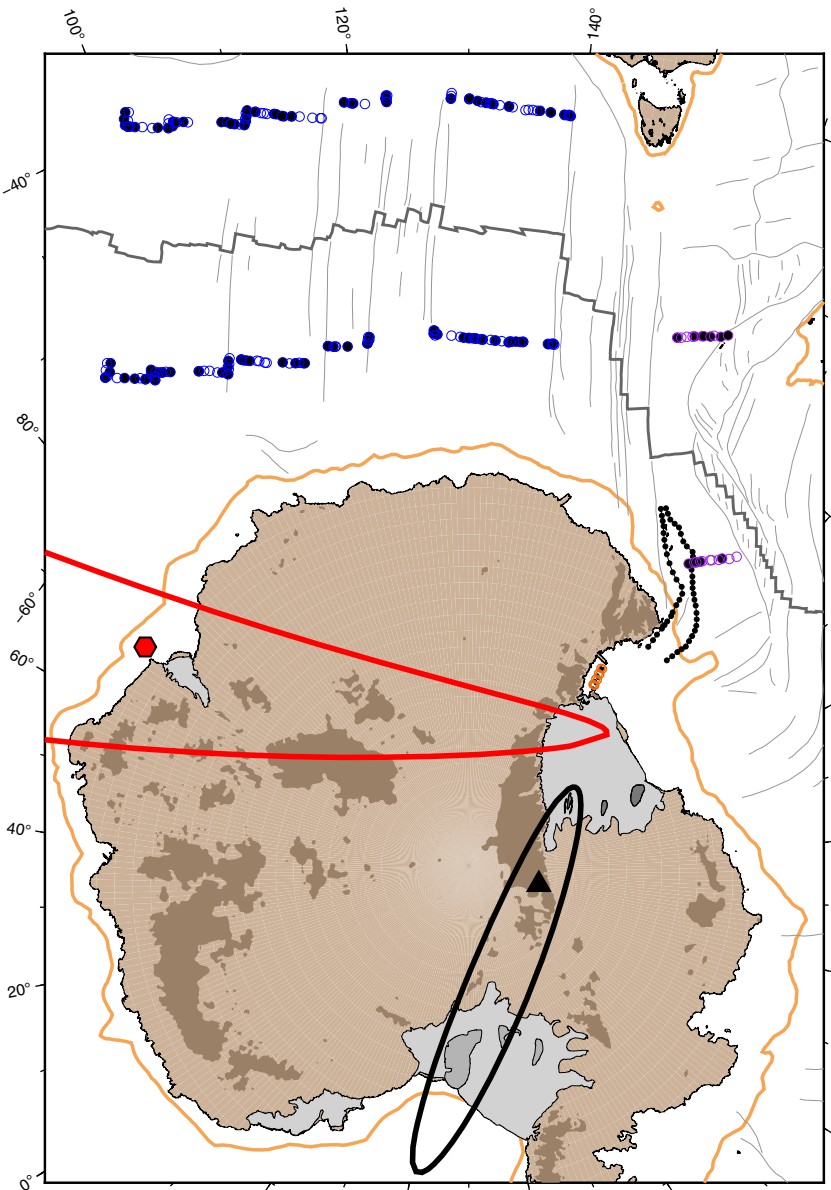

**Fig. 4** Data locations used to constrain the Australia-East Antarctica-West Antarctica three-plate solution for anomaly 8o time. Blue circles mark Australia-East Antarctica picks, purple circles mark Australia-West Antarctica and orange circles show the boundaries of the Terror Rift (East-West Antarctica picks). Empty and filled symbols are the original and rotated picks, respectively. The resulting pole is shown by the red hexagon and the Eocene-Oligocene stage rotation pole for East-West Antarctic motion (anomalies 18o–12o)[18] is shown by the black triangle. Ellipses delineate the 95% confidence boundaries for these poles. Orange lines delineate 1000 m contours. Black dotted lines delineate the northern edge of the WARS (Fig. 2)

contrasting lithospheric architectures observed along the rift and shoulders: extension in the north resulted in crustal thinning and flexural bending while significant convergence led to lithospheric foundering beneath the southern TAM, which was replaced at shallow depths by warm mantle material.

The kinematic evolution of the WARS influenced the thermal state of the lithosphere and therefore the distribution of volcanic activity. In the northern part of the rift system, the Adare Basin accommodated extension of 180 km between 43 and 26 Myr (rate of 12 mm yr$^{-1}$) and 35 km of total displacement between 26 and 11 Myr (2.3 mm yr$^{-1}$). Unlike the northern part of the rift system, where motions in the Neogene were almost an order of magnitude smaller than those during the Eocene-Oligocene, the predicted Neogene displacement in the southern part of the rift (34 km in the Bentley Subglacial Trough, 2.2 mm yr$^{-1}$) is comparable to that during the Eocene-Oligocene (61 km, 3.5 mm yr$^{-1}$). The mapped onland[9] and subglacial[41] Neogene volcanoes and distribution of high heat flows[5] are confined to the northern and southern parts of the rift system (Fig. 5). The Ross Ice Shelf region, where motion-oriented perpendicular to the front of the TAM and the rift axis during the Neogene was minimal, lacks such volcanic activity, although poor sounding coverage there may also contribute to this inference. The apparent link between the amount of motion and the volcanic activity may indicate that rift kinematics during the Neogene has governed and still governs[5] heat flow distribution. Motion in the Neogene was highest underneath the West Antarctic Ice Sheet, possibly triggering the formation of Neogene sub-ice basins and troughs[27,28] (e.g., Bentley Subglacial Trough, Fig. 5).

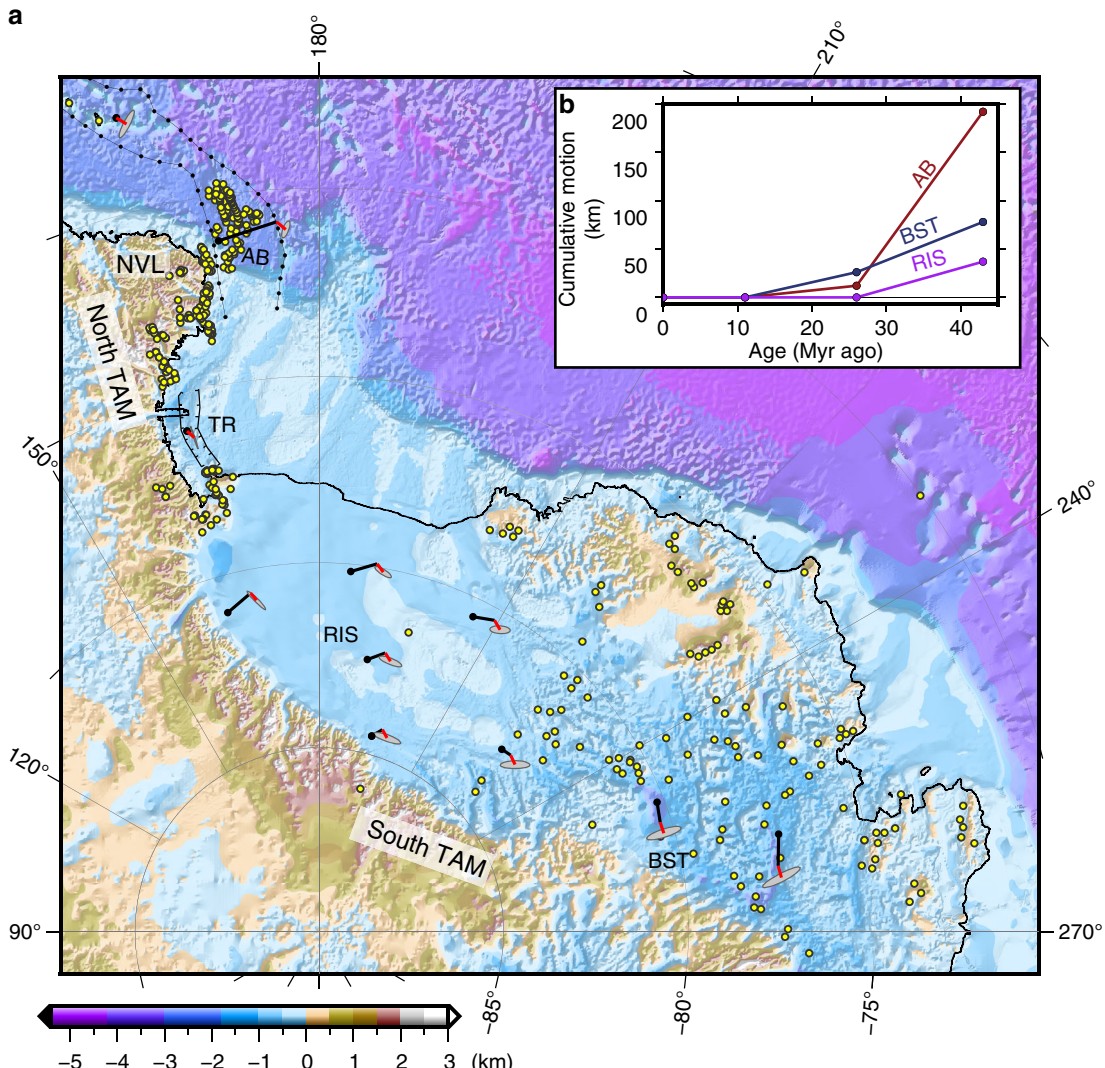

**Fig. 5** Predicted late Cenozoic displacements and their 95% confidence ellipses in the West Antarctic rift system. **a** Black lines show the displacements of East Antarctica relative to a fixed West Antarctica for the Eocene-Oligocene, calculated by extrapolating the stage pole 18o–12o[18] to cover the entire spreading period recorded by the Adare Basin (anomalies 20o–8o). Early Neogene displacements are shown by red lines with their 95% confidence ellipses. Only Neogene displacements are plotted in the locations of the Balleny Islands and Terror Rift. Yellow circles show mapped volcanoes[9,21,41]. Background map shows bedrock elevation and bathymetry[54]. **b** Cumulative motion perpendicular to the front of the TAM and the East-West Antarctic plate boundary. The Ross Ice Shelf (RIS) curve was calculated at location 84.5° S, 195.0° E. AB, Adare Basin; TR, Terror Rift; NVL, Northern Victoria Land; BST, Bentley Subglacial Trough

**Implications for global plate circuits**. The change in position of the East-West Antarctica Euler pole and reduction in rotation rate some 26 Myr ago correspond to a major reorganisation in the tectonics of the southwest Pacific. This period marks a change in the kinematics of the Pacific-Antarctic Ridge[42], the termination of seafloor spreading along the Pacific-Australia plate boundary[43] and the onset of oblique transform motion along the Macquarie Ridge[44] (Fig. 1). The cessation of motion between East and West Antarctica at around the time of anomaly 5o (11 Myr) has coincided with the transition of strike-slip-dominated faulting along the New Zealand Alpine Fault into oblique compression and continental collision[45,46]. At the same time, Pacific-Antarctic seafloor spreading started to undergo a major kinematic change[42] that was followed by up to 30° of clockwise change in spreading direction and a substantial increase in spreading rates. Interestingly, the evolution of the absolute motion of the Pacific Plate[47,48] coincided with the two Neogene East-West Antarctic kinematic events presented here.

A counterclockwise change in the direction of absolute plate motion happened between 30 and 24 Myr, at around the time when the Adare seafloor spreading ceased, and its absolute velocity began to increase sharply at around 11 Myr ago, about when rifting in Antarctica terminated.

The final cessation of the Antarctic rifting documented here entailed a simultaneous process whereby a major change in the direction of East-West Antarctic relative plate motions was coupled with a decrease in plate velocity. Similar observations have been documented in other divergent fossil boundaries[49,50], and therefore, they may be representative of a fundamental process that occurs during the final stages leading to cessation of a plate boundary.

Understanding the Neogene East-West Antarctic plate motions has, besides the regional consequences noted above, important implications for the tectonics of the southwest Pacific plate circuit as well as of the global plate circuit. Previous global plate tectonic studies have assumed that Antarctica was unified into a single

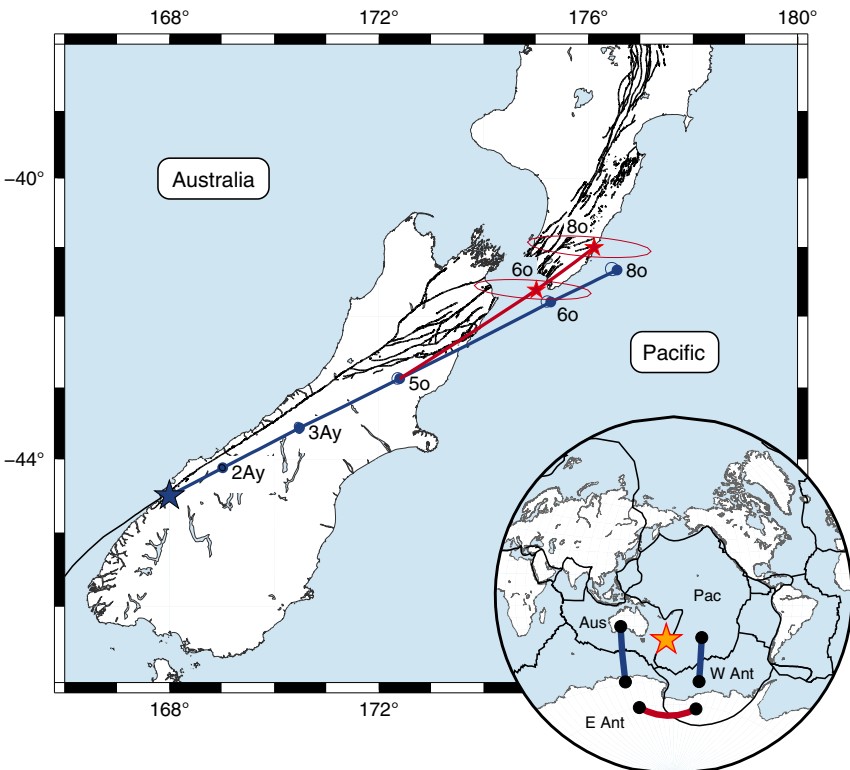

**Fig. 6** Pacific-Australia plate motions during the Neogene. Predicted plate motion trajectories for the Pacific Plate relative to the Australian Plate shown for a point along the New Zealand Alpine Fault. The blue line shows the trajectory calculated by the summation of Australia-Antarctic[30] and Antarctic-Pacific[42] motions, assuming that Antarctica was a single plate. Inclusion of East-West Antarctic motion between anomalies 8o and 5o (Table 1) results in the red trajectory for the early Neogene

plate prior to the Neogene, an assumption that led to inconsistencies between the resultant plate reconstructions and geological observations. The kinematics of the Alpine Fault in New Zealand, for instance, was determined by the summation of the Pacific-West Antarctica and East Antarctica-Australia rotations[30] (Fig. 6). The resultant reconstruction predicted that the direction of motion along the Alpine Fault remained nearly constant throughout the Neogene while geological observations suggested that a marked change in relative motion happened during the late Miocene[45,46]. Inclusion of the early Neogene East-West Antarctic motion in the plate circuit resulted in a late Miocene kinematic change from strike-slip-dominated to oblique-compressional Pacific-Australia plate motion (Fig. 6). The reconstructed position of a point at 26 Myr ago is shifted by 51 ± 20 km (1σ) to the northwest, indicating that the previously published estimate contains a bias for the early Neogene that is much larger than their formal uncertainties.

Neogene motions between East and West Antarctica also alter the predictions calculated by the global plate circuits. For instance, despite our substantially improved understanding of the Pacific-North America plate motions, a persistent discrepancy of up to ~100 km between the predicted and geological estimates of the accumulated offset along the margin still exists for the early Neogene[12,13] (Fig. 7). Using our new rotation, albeit with large uncertainty (±65 km of margin-parallel motion) stemming from the addition of a short plate boundary to the circuit, this gap is reduced by half for the early Neogene and eliminated for the last 16 Myr. Furthermore, the revised predicted plate motion trajectory for the Pacific Plate relative to the North America Plate extends seaward compared with the predictions computed with the assumption of a unified Antarctic Plate (Fig. 7). This difference (50 km in the direction oriented perpendicular to the

plate boundary at 20 Myr ago) suggests that there was less divergent motion across the plate boundary during the Neogene than previously estimated. Incorporating the Neogene kinematic evolution of East-West Antarctica in the global plate circuit would also affect other tectonic issues such as the motion of the Pacific Plate relative to the rest of the plates, the location of the predicted hotspot trails and the possible motion between the Indo-Atlantic and Pacific families of hotspots.

## Methods

**Poles of rotation.** We employ best-fitting criteria[35] and a statistical approach[36] to compute the rotation parameters and their uncertainties for a set of plate pairs. The overall motion of the Macquarie microplate since its creation at anomaly 3Ay relative to Australia was computed by calculating the rotation parameters of Macquarie relative to Antarctica based on data from the two corridors that comprise the microplate, the Tasman and Balleny corridors. We assigned 2 km uncertainty to the location of the well-navigated magnetic picks and 4 km uncertainty to the picks that originated from the archival, poorly navigated data. The locations of the relatively short offset Tasman fracture zone were assigned 4 km uncertainty and the locations of the long offset Balleny fracture zone were assigned 8 km uncertainty. Note that we obtain virtually the same solution regardless of which set of data we use, i.e., from the Balleny corridor and Balleny fracture zone and/or from the Tasman corridor and Tasman fracture zone, although the results obtained using the latter data set entail larger uncertainty. Finally, we summed the Macquarie-Antarctica solution with the rotation parameters of the anomaly 3Ay solution of the East Antarctica-Australia that was calculated based on data from the SEIR located west of the George V fracture zone[30] (Supplementary Fig. 2 and Table 1). Values of the statistical parameter $\hat{K}$, used to evaluate the assigned errors for the locations of the data points, were near 1, indicating that the uncertainties assigned to the data points were reasonable.

**Estimation of the Pacific-North America plate motion.** Observations for the geological offsets accommodated across the Pacific-North America plate boundary were taken from DeMets and Merkouriev[12] and references therein. This estimate, calculated by summing the deformation observed across the central Basin and Range and the central part of California, accumulates to 700 km of margin-parallel

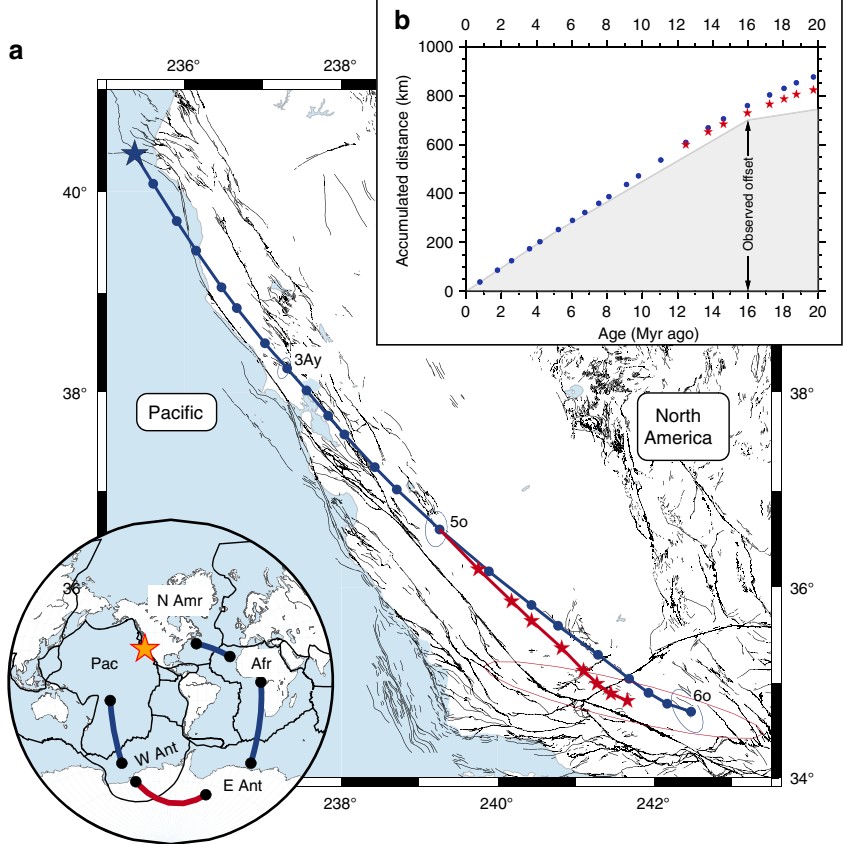

**Fig. 7** Pacific-North America plate motions during the Neogene. **a** Predicted plate motion trajectory for the Pacific Plate relative to the North America Plate. The blue line shows the trajectory calculated by the summation of the noise-reduced Pacific-Antarctica-Africa-North America plate circuit rotations[12], assuming that Antarctica was a single plate. Inclusion of East-West Antarctica motion between anomalies 8o and 5o (Table 1) results in the red trajectory. Ellipses show the resulting combined 95% uncertainty region for the reconstructed Mendocino Triple Junction (blue star). **b** Accumulated boundary-parallel displacement components (N40°W direction) calculated along the trajectories shown in panel **a** (blue and red stars were computed excluding and including, respectively, Neogene East-West Antarctica motion). Geological estimates of the boundary-parallel displacements across the plate boundary are shown by grey-shaded area (see Methods for details)

(N40°W) offset younger than 16 Myr. Although prior deformation is not well resolved, we added 45 km of deformation for the 20 to 16 Myr time period (Fig. 7) to account for early deformation in the Basin and Range and in faults located west of the San Andreas[51].

**Data availability**. The magnetic and fracture zone picks used to compute the kinematic models are available through the Global Seafloor Fabric and Magnetic Lineation (GSFML) repository[52].

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

## Acknowledgements
We thank the captains and crews of the *IB L'Astrolabe* for their dedicated work during the TACT01 and TACT02 cruises. We thank Charles DeMets for discussions, Florent Szitkar for his participation in the TACT02 cruise and IPEV for their support. R.G. was supported by Clare Hall College, University of Cambridge. All figures were created using GMT software. This is IPGP contribution 3955.

## Author contributions
Both authors designed and performed data acquisition. R.G. performed the kinematic analysis and wrote the paper with contributions from J.D.

## Additional information

**Competing interests:** The authors declare no competing interests.

