## [Peer Review File · Nature Communications]

REVIEWERS' COMMENTS:

Reviewer #1 (Remarks to the Author):

SUMMARY

The main result of this paper is that motion between East and West Antarctica did not stop abruptly at 26 Ma but continued until 11 Ma. As far as I know, this is the first time that this motion has been accounted for and has implications for global plate motion models. Importantly, it also decreases the propagation of errors in global plate circuits. Further, the authors suggest that the new plate kinematics can explain the volcanism or lack thereof and heatflow in the Antarctic rift system. I have no concerns about the methodology or soundness of the science. My main comments are about the way the results are presented.

MAJOR POINTS

Selling of the paper

For a Nature Communications paper, the authors need to do a better job at highlighting the importance of their work. In the introductory paragraph they state that "incorporation of this East-West Antarctica motion into the global plate circuit reduces the inconsistencies between the predicted and observed motions of the Pacific Plate relative to the Australian and North American Plates" but their results do far more than this. Some suggestions on ways to improve the paper to make it more relevant to a broader audience include:

1. What is the implication of their work on plate motion changes at ~11 Ma? Have the authors looked for kinks of Pacific hotspot tracks (e.g. Wessel et al. 2008)? Were there other changes in the Pacific around this time?
2. What is the implication of their work on the 27-23 Ma plate motion changes around the Pacific? Presumably, these plate motion changes were not caused by the abrupt cessation of E-W Antarctic motion but does it significantly alter the kinematics around the Pacific?
3. The Alpine Fault part of the story is impressive and convincing and it needs to be mentioned in the introductory paragraph and the Alpine fault figure moved to the main part of the manuscript
4. Line 39 should make its way into the opening paragraph
5. The section on the Cenozoic displacement fields is not entirely convincing and I do not think Fig. 4 needs to be in the main part of the manuscript. Looking at the displacement vectors in Fig. 4, I cannot follow that there is negligible displacement under the Ross Ice Shelf as the size of the vectors look similar in all three areas. Can the authors please clarify this? I am also not convinced that the kinematic evolution of the WARS has influenced the thermal state of the lithosphere. Could the lack of volcanism under the Ross Ice Shelf just be a function of it being covered in ice and therefore not mapped? I do note that the authors do state "although poor sounding coverage there may also contribute to this inference" but I do not think this is enough. They need to provide more evidence for this.

Figures

I have several suggestions regarding the figures.

1. A reconstruction figure is missing from the manuscript, perhaps one that looks like Figure 1 (which is a really nice figure) but at key time periods. It will make it a lot easier to follow which spreading system is discussed in the text and how everything fits together.
2. Figure 2 needs the Tasman and Bellamy corridors labelled and you need either an insert showing its location on a map or a box on Fig. 1. Do the positions of the red circles mean anything? If not, can you instead just use a line. If they mean something, tell us.
3. It is a bit ambiguous what the red dot and ellipse on Fig. 3 is. Please clarify. What are the orange lines?

4. As stated earlier, Figure 4 is a bit of a let down for the main text of the paper. I would move this figure to the supplementary section.
5. A lot of time is spent talking about the Alpine Fault in the text and the story here is really nice and convincing. I would move the Alpine Fault figure from the supplementary section to the main part of the manuscript.
6. Extended Data Fig. 1. From this figure, it is clear that several magnetic anomaly identifications can be made. So why were only a few chosen and why only provide poles at 8o and 3Ay?

Proof-read

This manuscript could have done with a thorough proof-read before submission. I have picked up some of the wording and grammatical errors (see minor points below) but I strongly suggest a thoroughly proof-read before its resubmission.

MINOR POINTS

Line 44: delete has

Line 52: delete "A" but explain a bit more about the type of faulting

Line 54: delete "a" and "been" and "the"

Line 83: which isochrons are we talking about here?

Line 116: delete "has"

Line 118: change to "the formation of which was previously"

Line 119: need to make a more balanced case for the origin of the Bellany Islands.

Line 194: where are the sub-basins?

Line 204-208: I think this needs to be reworded. It reads like a statement of the obvious. A decrease in velocity during the last stages of spreading. If they are talking about a novel process (?) then this needs to come out more with a better description. If not, perhaps the authors can delete these sentences.

Reviewer #2 (Remarks to the Author):

The authors have used seafloor magnetic anomalies to identify evidence for relative motion between East and West Antarctica between 26 and 11 million years ago. Previous reconstructions have considered these two plates to have been merged at 26 Ma – the age of the oldest magnetic anomaly evidence of their relative motion. This analysis shows relative motion between the plates continued until 11 Ma, from the increasing misfit of marine magnetic anomalies 6y and 8o and their conjugates when reconstructed assuming no motion between East and West Antarctica. The authors reconstruct poles of rotation to accommodate the newly identified relative motion and consider the ramifications for the global plate tectonic circuit. Two examples are presented, one from New Zealand and one from the West Coast of the USA.

The authors present the sense of motion implied across the Ross Sea and West Antarctic Rift and suggest that the newly identified pole of rotation could account for the different styles up uplift previously proposed along sectors of the Transantarctic mountains – associated with extension in the north and compression in the south with a transition in the central TAM.

The work presents a new result in a clear and accessible way. My expertise is limited to being one who would make use of these results, and so I cannot comment on the validity of the calculations represented in Table 1. Only the newly acquired data were shown in extended data, while the illustrated misfit between anomalies and conjugates is shown from the archival dataset. The source of the archival data was not cited at line 66/67 or in the Methods, and no archival data was shown in Extended Data. These should be shown in order to fully present the evidence. Within those limits I consider the findings to be well argued with appropriate acknowledgement of large uncertainties associated with the small relative motions being used to define rotation poles. The motions described are novel and likely to add to the understanding of tectonic motion within the region. The authors have considered the ramifications of their results across a broad region,

extending the significance of their findings.

A few suggestions to improve the clarity of the presentation:

Figure 1: show the location of figure 2 – figure 2 is rotated with respect to figure 1 and was a little hard to place on the first read.

Figure 2: Show the location of the tracks used to identify the magnetic anomalies being mapped, indicated the tracks illustrated in Extended Data figure 1.

Illustrate the archival data in Extended Data – similar to the present Extended Data Figure 1.

Reviewer #3 (Remarks to the Author):

The paper by Granot and Dymant evaluates the timing and magnitude of displacements between the East and West Antarctic blocks using robust kinematics arguments based on marine magnetic anomalies. They propose a surgical study of the anomalies located in the corridors between Balleny and Tasman fracture zones, partially corrected by the rotation of the Macquarie plate. East and West Antarctica limits are geologically marked by the West Antarctic Rift System (WARS) and the Trans-Antarctic Mountains (TAM).

Plate kinematics is important because it helps clarifying the geology of this region, still insufficiently understood and difficult to observe. The novelty is to show that the activity of this rift was not completed at 26 Ma, but at only 11 Million years. The associated deformation is also interpreted in term of plate tectonics and deformation for WARS, and tectonics and uplift consequences for TAM. Other consequences such as the relationships between volcanism, heat flow and extension are also discussed.

The quantification of this movement is important, not only for the understanding of the geology of this region, but also because it quantifies kinematically the movements between the border blocks of the Indian oceans and the Pacific Ocean. As an example, the relative motion between the hot spots of the two Pacific / Indo-Atlantic reservoirs will not be definitively resolved without a precise knowledge of the deformation zones or plate limits on the margins of the Pacific/Antarctic plates. This new set of data provides thus additional perspectives, albeit the motion inferred by their data on fig4 remains small.

The paper is well written, and along with the extended sheets, sufficiently informative. Although the presented figures are clear, I miss the presence of a small cartoon summarizing the tertiary evolution of the region with more structural information. This can be easily corrected by the authors.

The interesting scientific results, the quality of the data (along with the smart way of discussing them) and the overall good presentation make that this article must be published in Nature Communication.

Additional comments:

L 178 and/or elsewhere why to rule out the presence of a hot spot?

L111 "A strand of complex gravity features". insufficient. Please better describe.

Fig 4 Grey color is for ellipses: I guess volcanoes should be yellow? can we differentiate their ages?

Response to the reviewers' comments on the manuscript

We thank the referees for their supportive and constructive comments. Below we address the points raised in the reviews (our responses are shown in red).

Reviewer #1:

SUMMARY

The main result of this paper is that motion between East and West Antarctica did not stop abruptly at 26 Ma but continued until 11 Ma. As far as I know, this is the first time that this motion has been accounted for and has implications for global plate motion models. Importantly, it also decreases the propagation of errors in global plate circuits. Further, the authors suggest that the new plate kinematics can explain the volcanism or lack thereof and heatflow in the Antarctic rift system. I have no concerns about the methodology or soundness of the science. My main comments are about the way the results are presented.

MAJOR POINTS

Selling of the paper

For a Nature Communications paper, the authors need to do a better job at highlighting the importance of their work. In the introductory paragraph they state that “incorporation of this East-West Antarctica motion into the global plate circuit reduces the inconsistencies between the predicted and observed motions of the Pacific Plate relative to the Australian and North American Plates” but their results do far more than this. Some suggestions on ways to improve the paper to make it more relevant to a broader audience include:

1. What is the implication of their work on plate motion changes at ~11 Ma? Have the authors looked for kinks of Pacific hotspot tracks (e.g. Wessel et al. 2008)? Were there other changes in the Pacific around this time?

We thank the reviewer for raising these important points. Indeed, the implications of our work are wide and the results affect the global plate circuit linking the Pacific Plate with the rest of the world. Although we cannot, in the context of this manuscript, present all the implications of our work, we have modified and extended the discussion related to this topic and highlighted this point in the abstract. We have also moved the two supplementary figures that show the implications of our work on the plate circuits (Pacific-Australia and Pacific-North America), before and after 11 Ma, to the

main text. Obviously, there are more plate boundaries that can be looked at, but we chose to show those whose plate motion predictions have been significantly affected by our results and whose geology is relatively well understood. As such, they provide general and independent confirmation of our main results.

The locations of the Pacific hotspot tracks are not markedly altered by our results which led to only a minor change in the along-track ages of the 11-26 Ma part of the hotspot tracks. We thank the reviewer for pointing out the possible connection between our results and the Pacific absolute plate motion (WK08-A and WK08-D models, Wessel and Kroenke 2008 and Chandler et al., 2012, respectively). We added this detail to the discussion. Moreover, other kinematic changes around the Pacific that happened at ~11 Ma are now explicitly discussed.

2. What is the implication of their work on the 27-23 Ma plate motion changes around the Pacific? Presumably, these plate motion changes were not caused by the abrupt cessation of E-W Antarctic motion but does it significantly alter the kinematics around the Pacific?

The implications of the cessation of seafloor spreading in the Adare Basin at around 26 Ma on the kinematics associated with the Pacific Plate have already been thoroughly discussed by Cande et al., Nature, 2000. Our findings do not significantly change our understanding of this kinematic event, as we show that the motion after the cessation of seafloor spreading (i.e., between 26 and 11 Ma) was extremely slow, about an order of magnitude slower than the already slow motion that occurred between 43 and 26 Ma. To complete our discussion, we added a short description of the tectonic events that have taken place in the southwest Pacific during that 27-23 Ma time frame.

3. The Alpine Fault part of the story is impressive and convincing and it needs to be mentioned in the introductory paragraph and the Alpine fault figure moved to the main part of the manuscript

We thank the reviewer for this positive comment. As suggested, we moved the figure and added text to the discussion. Unfortunately, space constraints preclude us from explicitly referring to the Alpine Fault in the abstract.

4. Line 39 should make its way into the opening paragraph

We modified the abstract to include similar content.

5. The section on the Cenozoic displacement fields is not entirely convincing and I do not think Fig. 4 needs to be in the main part of the manuscript. Looking at the displacement vectors in Fig. 4, I cannot follow that there is negligible displacement under the Ross Ice Shelf as the size of the vectors look similar in all three areas. Can the authors please clarify this? I am also not convinced that the kinematic evolution of the WARS has influenced the thermal state of the lithosphere. Could the lack of volcanism under the Ross Ice Shelf just be a function of it being covered in ice and therefore not

mapped? I do note that the authors do state “although poor sounding coverage there may also contribute to this inference” but I do not think this is enough. They need to provide more evidence for this.

Figure 4 (now Figure 5) shows the displacement field within the WARS, computed based on the new (26-11 Ma) and existing (43-26 Ma) rotation poles. We believe that this kind of presentation, which has been used by previous kinematic studies (e.g., Cande et al., 2000; Muller et al., 2007; Granot et al., 2013), is essential: it converts the poles and their uncertainties, which are difficult to understand in terms of their kinematic consequences in time and space, into an accessible set of information that helps the reader better understand the evolution of the rift system.

Although the sizes of the vectors appear similar across the three areas, their directions are strikingly different. As a result, the extensional motion (i.e., the component of motion that is perpendicular to the axis of the rift) varies between the three areas, even when considering the uncertainties. We modified the text to clarify this point.

Regarding the volcanic activity, independent datasets show that the area of the Ross Ice Shelf is characterized by low heat-flow values compared with the heat flows shown for the West Antarctic Ice Sheet and for western Ross Sea areas (e.g., Martos et al., 2017 and Fisher et al., 2015). We also note that many volcanoes are mapped in the West Antarctic Ice Sheet area even though it is covered with ice. We therefore believe that the differences in mapped densities of volcanic activity between the three areas that constitute the rift system are generally real, despite the large uncertainties, particularly for the Ross Ice Shelf. Taken together, the findings thus strongly indicate that the volcanic density of the Ross Ice Shelf will differ from (be lower than) that of the rest of the rift.

Figures

I have several suggestions regarding the figures.

1. A reconstruction figure is missing from the manuscript, perhaps one that looks like Figure 1 (which is a really nice figure) but at key time periods. It will make it a lot easier to follow which spreading system is discussed in the text and how everything fits together.

We added a reconstruction map for 11 Ma (anomaly 5o) time, which is the new Figure 3.

2. Figure 2 needs the Tasman and Bellamy corridors labelled and you need either an insert showing its location on a map or a box on Fig. 1. Do the positions of the red circles mean anything? If not, can you instead just use a line. If they mean something, tell us.

We added the missing labels in Figure 2, and the location of the area shown in Figure 2 is now indicated in Figure 1. The red circles are used to highlight the edges of the rift, and therefore, their

exact positions have no significance. As the use of straight lines (without the red circles) would render these important boundaries less apparent, we kept the circles but reduced their sizes.

3. It is a bit ambiguous what the red dot and ellipse on Fig. 3 is. Please clarify. What are the orange lines?

We modified both Figure 3 (now Figure 4) and the caption to clarify these points.

4. As stated earlier, Figure 4 is a bit of a let down for the main text of the paper. I would move this figure to the supplementary section.

Please see our answer above concerning Figure 5. The figure is important for the discussion of the implications of our results for the West Antarctic rift system. We therefore feel that it should remain part of the main text.

5. A lot of time is spent talking about the Alpine Fault in the text and the story here is really nice and convincing. I would move the Alpine Fault figure from the supplementary section to the main part of the manuscript.

Done.

6. Extended Data Fig. 1. From this figure, it is clear that several magnetic anomaly identifications can be made. So why were only a few chosen and why only provide poles at 80 and 3Ay?

Only the 80 pole was computed for the East-West Antarctic plate motion because the temporal details of the extension in the Terror Rift are insufficient to use them as a basis to divide the period (26-11 Ma) into several stages. Due to the absence of East-West Antarctic motion after 11 Ma, we did not interpret younger anomalies, besides anomaly 3Ay which was needed for the reconstruction of the Macquarie microplate. We added a short explanation about this point.

Proof-read

This manuscript could have done with a thorough proof-read before submission. I have picked up some of the wording and grammatical errors (see minor points below) but I strongly suggest a thoroughly proof-read before its resubmission.

Done.

MINOR POINTS

Line 44: delete has

Done.

Line 52: delete “A” but explain a bit more about the type of faulting

We added a short description of the nature of the lower and middle Miocene faulting event.

Line 54: delete “a” and “been” and “the”

Done.

Line 83: which isochrons are we talking about here?

We added the name of the isochrons.

Line 116: delete “has”

Done.

Line 118: change to “the formation of which was previously”

Done.

Line 119: need to make a more balanced case for the origin of the Bellany Islands.

We modified the sentence to better emphasize that the origin of the Balleny Islands is not entirely clear.

Line 194: where are the sub-basins?

We added text and a reference to Figure 5 to clarify this point.

Line 204-208: I think this needs to be reworded. It reads like a statement of the obvious. A decrease in velocity during the last stages of spreading. If they are talking about a novel process (?) then this needs to come out more with a better description. If not, perhaps the authors can delete these sentences.

This sentence highlights the observation that the termination of rifting in Antarctica followed a combined process: a major change in the direction of relative plate motion and a decrease in plate velocity. We modified the text to make it clear.

Reviewer #2 (Remarks to the Author):

The authors have used seafloor magnetic anomalies to identify evidence for relative motion between East and West Antarctica between 26 and 11 million years ago. Previous reconstructions have considered these two plates to have been merged at 26 Ma – the age of the oldest magnetic anomaly evidence of their relative motion. This analysis shows relative motion between the plates continued until 11 Ma, from the increasing misfit of marine magnetic anomalies 6y and 8o and their conjugates when reconstructed assuming no motion between East and West Antarctica.

The authors reconstruct poles of rotation to accommodate the newly identified relative motion and consider the ramifications for the global plate tectonic circuit. Two examples are presented, one from New Zealand and one from the West Coast of the USA.

The authors present the sense of motion implied across the Ross Sea and West Antarctic Rift and suggest that the newly identified pole of rotation could account for the different styles up uplift previously proposed along sectors of the Transantarctic mountains – associated with extension in the north and compression in the south with a transition in the central TAM.

The work presents a new result in a clear and accessible way. My expertise is limited to being one who would make use of these results, and so I cannot comment on the validity of the calculations represented in Table 1. Only the newly acquired data were shown in extended data, while the illustrated misfit between anomalies and conjugates is shown from the archival dataset. The source of the archival data was not cited at line 66/67 or in the Methods, and no archival data was shown in Extended Data. These should be shown in order to fully present the evidence. Within those limits I consider the findings to be well argued with appropriate acknowledgement of large uncertainties associated with the small relative motions being used to define rotation poles. The motions described

are novel and likely to add to the understanding of tectonic motion within the region. The authors have considered the ramifications of their results across a broad region, extending the significance of their findings.

We thank the reviewer for the constructive and supportive review. We created a new supplementary figure (Supplementary Figure 3) showing the archival magnetic data that were used to construct the location of the isochrons shown in Figure 2 of the main text. We also added text to the figure caption to highlight the source of the archival data.

A few suggestions to improve the clarity of the presentation:

Figure 1: show the location of figure 2 – figure 2 is rotated with respect to figure 1 and was a little hard to place on the first read.

We added the location of Figure 2 in Figure 1 and modified the figure caption accordingly.

Figure 2: Show the location of the tracks used to identify the magnetic anomalies being mapped, indicated the tracks illustrated in Extended Data figure 1. Illustrate the archival data in Extended Data – similar to the present Extended Data Figure 1.

We added a new supplementary figure (Supplementary Figure 3) that shows all of the data (archival and new) that were used to map the isochrons in the Balleny and Tasman corridors as shown in Figure 2.

Reviewer #3 (Remarks to the Author):

The paper by Granot and Dymant evaluates the timing and magnitude of displacements between the East and West Antarctic blocks using robust kinematics arguments based on marine magnetic anomalies. They propose a surgical study of the anomalies located in the corridors between Balleny and Tasman fracture zones, partially corrected by the rotation of the Macquarie plate.

East and West Antarctica limits are geologically marked by the West Antarctic Rift System (WARS) and the Trans-Antarctic Mountains (TAM).

Plate kinematics is important because it helps clarifying the geology of this region, still insufficiently understood and difficult to observe. The novelty is to show that the activity of this rift was not completed at 26 Ma, but at only 11 Million years. The associated deformation is also interpreted in term of plate tectonics and deformation for WARS, and tectonics and uplift consequences for TAM. Other consequences such as the relationships between volcanism, heat flow and extension are also

discussed.

The quantification of this movement is important, not only for the understanding of the geology of this region, but also because it quantifies kinematically the movements between the border blocks of the Indian oceans and the Pacific Ocean. As an example, the relative motion between the hot spots of the two Pacific / Indo-Atlantic reservoirs will not be definitively resolved without a precise knowledge of the deformation zones or plate limits on the margins of the Pacific/Antarctic plates. This new set of data provides thus additional perspectives, albeit the motion inferred by their data on fig4 remains small.

The paper is well written, and along with the extended sheets, sufficiently informative. Although the presented figures are clear, I miss the presence of a small cartoon summarizing the tertiary evolution of the region with more structural information. This can be easily corrected by the authors. The interesting scientific results, the quality of the data (along with the smart way of discussing them) and the overall good presentation make that this article must be published in Nature Communication.

We thank the reviewer for the helpful and supportive review. The focus of this work is the kinematic evolution of the rift system and the consequences of that evolution on the regional and global scales. We believe that adding another figure and the corresponding text to discuss the tertiary evolution of the rift system, beyond what is already discussed as general background, will make the manuscript more difficult to read and change its focus.

Additional comments:

L 178 and/or elsewhere why to rule out the presence of a hot spot?

Based on a comment from Reviewer #1, we modified the sentence in Line 119 to better emphasize the lack of clarity surrounding the origin of the Balleny Islands.

L111 “A strand of complex gravity features”. insufficient. Please better describe.

We expanded the description to clarify this point.

Fig 4 Grey color is for ellipses: I guess volcanoes should be yellow? can we differentiate their ages?

We changed the color signifying volcanoes to yellow. The ages of all sub-ice volcanoes (de Vries et al., 2017) are only crudely estimated, and while many of the on-land volcanic bodies are isotopically dated, no pattern for the ages is apparent, at least not one that emerges from the available data. We note, however, that all of these volcanic bodies seem to have formed during the Neogene.